# Ehlers-Danlos: A Literature Review and Case Report in a Colombian Woman with Multiple Comorbidities

**DOI:** 10.3390/genes13112118

**Published:** 2022-11-15

**Authors:** María José Fajardo-Jiménez, Johanna A. Tejada-Moreno, Alejandro Mejía-García, Andrés Villegas-Lanau, Wildeman Zapata-Builes, Jorge E. Restrepo, Gina P. Cuartas, Juan C. Hernandez

**Affiliations:** 1Centro Universitario de Ciencias de la Salud, Universidad de Guadalajara, Guadalajara 44350, Mexico; 2Grupo Genética Molecular GENMOL, Universidad de Antioquia UdeA, Medellín 050010, Colombia; 3Grupo Neurociencias de Antioquia GNA, Universidad de Antioquia UdeA, Medellín 050010, Colombia; 4Infettare, Facultad de Medicina, Universidad Cooperativa de Colombia, Medellín 050012, Colombia; 5Grupo OBSERVATOS, Facultad de Educación y Ciencias Sociales, Tecnológico de Antioquia—Institución Universitaria, Medellín 050034, Colombia; 6Grupo Neurociencia y Cognición, Facultad de Psicología, Universidad Cooperativa de Colombia, Medellín 050012, Colombia

**Keywords:** Ehlers-Danlos syndromes, genetic diseases, collagen, neuropsychological assessment, case report

## Abstract

Ehlers-Danlos syndromes (EDS) are a heterogeneous group of genetically transmitted connective tissue disorders that directly affect collagen synthesis, with a broad range of symptoms. Case presentation: This study presents a clinical case of a Colombian woman with myopathic EDS and multiple comorbidities taking 40 years of medical history to make the right diagnosis. This article also presents a review of the current literature on EDS, not only to remind the syndrome but also to help the clinician correctly identify symptoms of this diverse syndrome. Conclusion: A multidisciplinary approach to the diagnosis of the patient, including clinical and molecular analysis, and neuropsychological and psychological assessment, is important to improve the treatment choice and the outcome prediction of the patients.

## 1. Introduction

The Ehlers-Danlos syndromes (EDS) are a heterogeneous group of inherited connective tissue disorders. They are clinically characterized by skin fragility, skin hyperextensibility, joint hypermobility, and excessive bruising [1]. Based on the literature, 1/5000 people live with EDS classical (types 1 and 2) and with EDS hypermobility (type 4) in almost 80% of the population diagnosed with EDS [2]. However, others consider EDS hypermobile type (hEDS) to be the most common, with a prevalence of 80–90% [3]. Recent studies have recognized 13 EDS subtypes. In the past years, the clinical and genetic heterogeneity of this condition was studied. Hence, now it is known that these disorders differ from one another [4] and can often be diagnosed based on family history and clinical criteria, including the degree and nature of involvement of skin, joints, skeleton, and vasculature [5]. The clinical spectrum varies from mild skin and joint hyperlaxity to severe physical disability and life-threatening vascular complications [6]. Genetic testing for most types of EDS has been defined and can be useful for diagnosis. Thus, the clinical manifestations, pathogenesis, diagnosis, management, and complications are different due to the diversity in the presentations of this disease. In some patients, neurological and spinal manifestations have been reported [7]. Furthermore, anxious symptomatology, anxiety disorders [8], depressive symptoms, and anxious obsessive/compulsive personality disorder [9] have also been found in a few patients. There are few studies on neuropsychological functioning. However, the findings suggest deficits in concentration [10] and cognitive impairment in visuospatial problem-solving, attention, and memory [11]. 

### 1.1. Clinical Manifestations and Pathogenesis

EDS, the so-called disorder of many faces [6], occurs due to the different alterations involved in the synthesis of encoding fibrillar collagens or collagen-modifying enzymes [2]. However, recent studies have demonstrated the implication of other extracellular matrix (ECM) molecules such as proteoglycans and tenascin-X or genetic defects in molecules involved in intracellular trafficking, secretion, and assembly of ECM proteins [6]. Other studies demonstrated that mutations in COL5A1 and COL5A2 induce a reduction in the synthesis of type V collagen, demonstrating their central role in the pathogenesis of classical EDS [6,12]. Type V collagen is a small quantity of fibrillar collagen extensively present in a variety of tissues, which is primarily found in two particular ways: heterotrimers ([a1(V)]2a2(V) and a1(V)a2 (V)a3(V)) or as a1(V)3 homotrimers [13]. The most common collagen in tissues is a1(V)2a2(V), which is mostly found in the cornea [14,15]. Types I and V collagen gather, forming heterotypic collagen fibrils. Therefore, COL5A1 haploinsufficiency creates a boundary in the satisfactory production of heterotrimers, making the structure assembly fragile. The two fibrillar collagens connect to form fibrils requiring the NH2-terminal domain of type V collagen to project to the surface. This domain reaches an unfavorable concentration, making the assembly and regulation of the fibril diameter less favorable [15]. Diversity in EDS is due to different mutations in COL5A1 and COL5A2 as well as in classical-like EDS due to tenascin-X deficiency (OMIM #606408) [16]; cardiac-valvular EDS for a mutation in type I collagen (COL1A2) (OMIM #225320) [17]; vascular EDS (OMIM #130050) due to mutations in type III procollagen (most of them in the COL3A1 gene) [18]; and kyphoscoliosis EDS (OMIM #225400) due to mutations in PLOD1, resulting in lysyl hydroxylase deficiency [19]. Arthrochalasia (OMIM #130060) is caused by a loss of exon 6 in either COL1A1 (EDS VIIA) or COL1A2 (EDS VIIB), leading to structural defects in type I collagen [20,21], whereas dermatosparaxis (OMIM #225410) occurs due to mutations in the ADAMTS2 gene, leading to a deficiency of procollagen I N-terminal peptidase [22].

Diversity in clinical manifestations has been implicated in every presenting form of this disease. The proper manifestations are multi-systemic, with the following most common features [6]: 

Skin hyperextensibility: It is positive when it can be stretched over 3 cm or more at a neutral site, such as the neck, the distal part of the forearm, the dorsum of hands, the elbow, or knees, until resistance can be felt [23] due to COL5A1 haploinsufficiency [24], COL5A1, and COL5A2 [25].

Joint hypermobility: The joint laxity and hypermobility, involving both proximal and distal joints, is a major criterion in most of the EDS forms and is evaluated by the Beighton Hypermobility Score [26] (Table 1). This alteration might be associated with tenascin-X [27].

Tissue fragility: The skin fissures easily after minor trauma, more likely over pressure points and exposed areas, presenting as thin and wide atrophic scars often referred to as “cigarette paper scars” [6] that might be resulting from the alteration of decorin, fibromodulin, and lumican [28,29,30,31].

### 1.2. Classification 

A study of 126 suspected classical EDS patients revealed that 93 out of the 102 that demonstrated all three major Villefranche criteria for classical EDS, such as skin hyperextensibility, widened atrophic scars, and joint hypermobility, were associated with type V collagen mutations (90% of their population) [25]. However, this did not reflect the hypermobile type (hEDS), which is the most common, but only the classic type [5].

The major and minor clinical diagnostic criteria are different because of the clinical variations seen in each subtype. The diagnostician would prefer to make a molecular diagnosis. However, the molecular pathogenesis reviewed in the literature till today is useful but not enough [32]. The clinical presentation of the 13 subtypes is presented in Table 2. 

The Villefranche classification nomenclature (1997) has facilitated the diagnosis and classification of EDS [5]. A new international classification of the EDS emerged 20 years later, replacing the older nomenclature and presenting a new scheme for the 13 types of EDS (Table 2) [4]. 

### 1.3. Management, Prognosis, and Complications

Health professionals are advised to refer any potential patient with EDS or at least once the diagnosis is suspected [33] to a multidisciplinary health team. In patients, due to the multiple manifestations, a medical and biopsychosocial treatment program is the best treatment option [34]. 

There is no cure for EDS. The treatment focuses on managing the symptoms and preventing complications through physiotherapy and pain medication [35]. Pharmacological therapy is limited and lacks evidence [36]. Unfortunately, the incidence of pain in this syndrome could reach 100% [37] in hEDS, and the prevalence of chronic pain could reach 90% in patients with different types of EDS [38]. The key management cannot be generalized but is focused on every patient and type of EDS, mostly involving dermatological, musculoskeletal, cardiovascular, and ophthalmological features. As a result, a few specific management guidelines have been developed. In classical EDS, due to COL1A1, the central point of management is the cardiovascular system, involving the measurement of aortic root size, echocardiogram, and blood pressure [39]. It also requires a particular focus on the musculoskeletal system [23], whereas classical-like EDS, due to tenascin-X deficiency (clEDS) and dermatosparaxis (dEDS), does not have specific management criteria. 

EDS (cvEDS) is similar to classic EDS, with more rigorous studies for heart function. Arthrochalasia focuses particularly on the musculoskeletal and dermatological systems, such as the management of orthopedic problems, especially dislocations. However, in this case, the primary management should be during pregnancy, as women might be susceptible to tearing of the perineal skin and having a postpartum extension of episiotomy incisions and prolapse of the uterus and/or bladder. Brittle cornea syndrome (BCS) affects the heart of the ocular system, with primary prevention of corneal rupture and the auditive system [39]. Musculocontractural (mcEDS) pays special attention to musculoskeletal, cutaneous, cardiovascular, visceral, and ocular complications [40,41]. For the rest of the subtypes, the management is non-specific. Hence, it is recommended to focus on avoiding complications and improving the symptoms as they appear. (In this case, the subtypes are: vascular EDS, hypermobile EDS, dermatosparaxis EDS, kyphoscoliotic EDS, spondylodysplastic EDS, myopathic EDS, and periodontal EDS [39]).

The patient’s complaints of pain generally confine to the knees and thighs [42]. However, it is not the only clinical presentation that makes chronic pain an expression of EDS. Further evidence-based studies should be undertaken to manage the symptoms of pain, which require an interdisciplinary approach. If the pain is of an inflammatory genesis, then it can be managed pharmacologically with non-steroidal anti-inflammatory drugs (NSAIDs). Moreover, opioids can be used for a brief period. Furthermore, acetaminophen is recommended in cases of adverse reactions or contraindications associated with NSAIDs [43]. At specific painful points, topical lidocaine can be used for subluxations and injections at 1% concentration [44,45]. Pain control measures can also include massage therapy, splints or braces, and heat therapy [46].

The prognosis and complications vary from one form to the other. The severity of clinical pain, difficulties, or dislocations is proportional to the intensity of activities in every patient. The classical and hypermobility types relatively undergo normal aging, whereas the vascular subtype, which is the most dangerous of all, has a propensity to undergo arterial rupture and hollowing of organs by the age of 40 years [6,26]. Referral to specialists is encouraged in patients with EDS for accurate classification and effective management. 

## 2. Case Presentation

In this study, a 46-years-old female patient diagnosed with EDS, born in Colombia with a 40-year history of several clinical conditions, and suffering from chronic disabling pain for more than eight years has been presented. The patient was referred to the Research Biomedical Laboratory at the Universidad Cooperativa de Colombia, Medellín, Colombia. The patient worked as an assistant in a clinical laboratory of the nursery. At the birth of the index case, the father and mother were 38 and 22 years old, respectively. The parents were not blood relatives and were healthy (Figure 1). 

The current condition began at age six with spontaneous mandibular dislocation, costochondritis, hypermobility of joints, irritable bowel syndrome, hemorrhoids, and continuous tachycardia (some of the clinical manifestations, like spontaneous mandibular dislocation, hyperlaxity of joints, and tachycardia, continued throughout her growth). When the patient was eight years old, she presented with nasal and vaginal hemorrhages without an apparent reason and had severe pelvic and lower limb varicose veins. When she was 38 years old, she presented with mild lumbar osteoarthritis, scoliosis, bilateral renal lithiasis (diagnosis by ultrasonography), loss of bilateral patellofemoral space compatible with chondromalacia (diagnosed with anteroposterior and lateral comparative X-rays), incipient spondylosis in L4 (diagnosed with magnetic resonance imaging (MRI) of Lumbrosacral Simple Column), uterine myomatosis, five months of menometrorrhagia (treated with medroxyprogesterone) hearing loss, vertigo, imbalance, and rhinosinusitis. Currently, she presented with involuntary movements in the upper extremities (UE) and lower extremities (LE), dry eye syndrome, allergic rhinitis, edema in the morning and occasionally at night in LE, gastroesophageal reflux disease (diagnosed by upper endoscopy and biopsy), vertigo, disabling headache, and glaucoma. On physical examination, hyperextensibility of the skin (Figure 2A), keloid atrophic scarring, generalized hypermobility of the joint (Figure 2B), dislocations of the jaw and elbow, easy skin bruisability, spontaneous bruises, striae in the LE and around the pelvic area, abdominal hernia (umbilical type), dental crowding, mandibular deviation to the left (Figure 2C), arachnodactyly, bruisability, muscular hypotonia, kyphoscoliosis, central Webber, Rinne+, lower and upper limb bowing, momentary contractures and involuntary hand movements were noted. 

The clinical examination revealed different clinical signs, like positive Raynaud’s phenomenon, Steinberg+, and Romberg+. She scored six on a Beighton scale out of nine (Table 2), whereas five on the Five-Point Questionnaire (Table 3), which is a quick tool for assessment when generalized joint hypermobility is suspected in the patient. 

### 2.1. Neurological and Psychological Assessment 

Neuropsi (Figure 3), a standardized neuropsychological test, was used for assessing executive functioning, memory, and attention [47]. The standardized score for attention and executive functions was 63; for memory it was 65; and for attention and memory, it was 62. These three neuropsychological functions revealed severe alterations as per age and schooling. Anxiety and depression were assessed using Beck Anxiety Inventory [48] and Beck Depression Inventory [49]. Anxiety with an 8 score was at the intermediate level, and depression with a 3 score was at a minimum level. The level of stress, assessed through the stress assessment scale [50], was very low (29 scores). The personality analysis revealed [51] low levels of neuroticism (pth 1), high level of consciousness (pth 95), and normal levels of openness to experience (pth 30), extraversion (pth 50), and agreeableness (pth 65).

The laboratory tests’ history revealed different findings: The electrocardiogram (ECG) showed blockage of the left branch, the MRI of the temporomandibular joint (TMJ) reported temporomandibular arthrosis changes, especially, with alteration of the condylar morphology, disarrangement consisting of a non-reducible right anterolateral dislocation, and a left anterior reducible dislocation. A simple bilateral ATM MRI reported temporomandibular arthrosis with flattening and sclerosis of the condyle, enlargement of the glenoid fossa, and sclerosis of the articular temporal eminence, with decreased anterosuperior articular space in the right TMJ, whereas no apparent alteration was observed in the left TMJ. The disc had lost its biconcave configuration and was dislocated anteriorly and laterally, without being recovered in the dynamic sequences. Hence a limitation in the buccal opening, and the the condyle without reaching the corresponding joint temporal eminence were evidenced. The venous duplex showed insufficiency of the greater saphenous vein in the distal third of the thigh. The ultrasonography venous duplex: LE revealed an incipient varicose vein of left pelvic origin and tributary of the long saphenous vein with a perforator in the leg was seen. The antinuclear antibodies (ANAS) were negative. A simple and contrasted brain MRI revealed a mild frontal subcortical microangiopathy, cyst of the right choroidal fissure, right vertebral hypoplasia, mucous retention cysts in the frontal sinus, mild decrease in the anteroposterior diameter of the ocular globules with a normal Holter. 

Additionally, to evaluate the association of the neuropsychological state of this patient with physiological functions; the serotonin, tryptophan, catecholamines (epinephrine, norepinephrine, and dopamine), and cortisol were determined in serum samples in a reference laboratory from Medellín, Colombia. The values obtained were normal for each parameter; serotonin: 119.8 ug/L, tryptophan: 27.4 µmol/L, epinephrine: less than 5 pg/mL, norepinephrine: 203.1 pg/mL and dopamine: less than 30 pg/mL. However, cortisol levels were lower than expected (3 ug/dL; reference range: 3.7–19.4 ug/dL). 

These neurobiological findings have been associated with the emotional and personality profiles of the patient. Clinical and psychometric evidence revealed a decrease in affect display (emotional blunting). Low cortisol levels have been associated with anti-arousal disengagement defense mechanisms principally involving emotional numbing and “shame-laden depression” [52]. The possibility that this profile is a consequence of social stress (personal and social maladjustment, low self-esteem, and poor self-image) generated due to the disease history cannot be excluded. Some of the behavior patterns of the patient suggested that she could be feeling ashamed due to her medical condition. This psychosocial factor should be considered during the treatment.

### 2.2. Neurological Examination 

No personal history of psychopathology was present. However, a family history of a psychological disorder was present. There was a history of traumatic brain injury with loss of consciousness for three days at 23 years. However, no alteration of the cranial nerves was noted. A few osteotendinous reflexes showed hyperreflexia and hyporeflexia. No motor or sensory abnormalities were present. For three years, the clinical features of migraine with a sensation of “head dullness”, a feeling of pressure, and stabbing pain accompanied by photophobia that increased with the consumption of sweets were present. Alterations in the higher mental functions or behaviors were not seen.

### 2.3. Genetic Study

The coding regions of the genome were sequenced by Next Generation Sequencing (NGS) in Macrogen, Inc. ^TM^ (Seoul, Republic of Korea). The samples were prepared based on the Agilent Sure Select Target Enrichment Kit preparation guide and the libraries were sequenced with an Illumina platform sequencer (2 × 10^1^ base pair paired-end reads) following Illumina platform protocol. Finally, the data was processed by the software HCS (HiSeq Control Software v2.2.68 Illumina, Inc., San Diego, CA, USA) to obtain the raw information. The data product of the sequencing was converted to the format FASTQ using the package Illumina bcl2fastq. The quality of reads was evaluated with the fastqc_v0.11.5 tool of the Babraham Institute (https://www.bioinformatics.babraham.ac.uk/projects/fastqc (accessed on 6 July 2021)). Then, reads were mapped against the reference human genome (UCSC hg19) using the Burrows-Wheeler Aligner tool, bwa-0.7.12 (http://bio-bwa.sourceforge.net/bwa.shtml (accessed on 6 July 2021)). Variant calling was performed following the Best Practices for Germline SNP and amp; Indel Discovery in Whole Genome and Exome Sequence Protocol of the Broad Institute’s Genome Analysis ToolKit, GATK v3.4.0, (https://software.broadinstitute.org/gatk (accessed on 6 July 2021)). Finally, the variants were annotated with the SnpEff tool v4.1g (http://snpeff.sourceforge.net/SnpEff.html (accessed on 6 July 2021)) and wAnnovar (http://wannovar.wglab.org (accessed on 6 July 2021)).

A possible pathogenic rare variant, c.C7853T; p.T2618M, in the COL12A1 gene was identified. This variant was known as rs201988277 by the dbSNP database of the National Center for Biotechnology Information (NCBI) [53]. It is a non-synonymous, rare variant located in exon 36 of the gene with a frequency of the minor allele (MAF) less than 0.001 in the general population based on the 1000 Genomes Project (1000G) [54], the Exome Aggregation Consortium (ExAC) [55], and the Genome Aggregation database (genomeAD) [56]. Additionally, it is considered deleterious by more than three pathogenicity predictors, including Sorting Intolerant From Tolerant (SIFT) [57], Polymorphism Phenotyping v2 (PolyPhen2) [58], Protein Variation Effect Analyzer (PROVEAN) [59], and Combined Annotation Dependent Depletion (CADD) [60] (CADD score above 20 is considered as pathogenic) (Table 4 and Appendix A). Further, this variant is considered as pathogenic moderate (PM1) because it is located in a mutational hot spot and/or on a critical and well-established functional domain (e.g., the active site of an enzyme). The variant is absent from controls in the 1000 Genomes, Exome Sequencing Project, and other allele frequency databases (PM2) without benign variation. Additionally, a reputable source reported this variant to be pathogenic. However, the information was not available for confirmation (PP5) based on the Wintervar ACMG Classification [61,62]. Moreover, Varsome provided additional information, stating that the variant was a missense one in a gene with a low rate of benign missense variants (PP2) and added pathogenic supporting (PP3) because of several lines of computational evidence supporting a deleterious effect on the gene or gene product (conservation, evolutionary, splicing impact, etc.). If this information is added manually to Wintervar, the result would likely be a pathogenic variant (because two features between PM1–PM6 and two between PP1–PP5 would likely give a pathogenic variant) [61] (Appendix A).

## 3. Discussion

The heterogeneity in EDS makes it hard to make a specific diagnosis of the subtype only with symptoms. Hence, patients have notified a delay of 14 years on average in making an accurate diagnosis [63]. In this patient, the specific diagnosis was made when she was 38 years old. This delay worsened the symptoms and the prognosis. The initial diagnosis was joint hypermobility syndrome (JHS) that has overlap with EDS [64]. Those are patients who vividly remember contorting their bodies into strange shapes when asked by other children [65]. It is now known that the first manifestations of this syndrome start in the principal years of life. Hence, this syndrome is associated with prematurity [66]. In this study, the patient who is herein presented was also born at 32 weeks. JHS patients have clinical manifestations that are suggestive of fibromyalgia. Hence, they are commonly misdiagnosed [67], similar to the case of our patient. The clinical manifestations in every patient with EDS continue growing with them, the natural history of children with hEDS reveals high levels of pain and fatigue [68], which was the third diagnostic criteria for the patient who had joint complaints at six years of age. Moreover, they could be confused with rheumatic diseases, ignoring the complexity of the syndrome [69]. 

Likewise, in this patient, generalized hyperalgesia was frequently present in hEDS, correlating with the symptoms revealed by the patient from a young age. Gastrointestinal implications are recurrent in patients with EDS, presenting as unexplained gastrointestinal symptoms, abdominal emergencies, and hernias, among others [70,71,72]. This patient had an umbilical hernia. She also had gastroesophageal reflux disease, which was confirmed by biopsy. A connection between the fragility of the capillaries, easy bruising, and bleeding has been previously made [73], thus explaining the nasal bleeding observed in this patient. Previous studies have noted changes in collagen mass or configuration of the divergent collagen subtypes in patients with varicose veins [74,75,76]. Early-onset varicose veins are associated with the vEDS [2]. However, in this patient, varicose veins were seen in the vulva and LE. Varicose veins are more common among pregnant women with EDS-HT [77], thus suggesting the presence of a convergent pathophysiological pathway [78]. Lumbar spondylosis is also noted in these patients [79]. This results in mechanical pain in the patient [7]. In this patient, on MRI, incipient spondylosis was noted in L4, because of which the patient has been currently using a walking stick. Generalized joint laxity has been associated with chondromalacia [80], patellofemoral and tibiofemoral kinematics [81], and also with patellofemoral pain [82]. This patient presented X-rays of comparative AP and lateral knees due to loss of bilateral femoropatellar space compatible with chondromalacia. Additionally, she also had complaints of chronic pain in the knee. Hearing loss is common in patients with EDS [83], as was observed in this lady. Patients with EDS report different atopic clinical manifestations [84]. Our patient also presented with allergic rhinitis. Patients often are misdiagnosed by underestimating symptoms in association with psychiatric and psychological manifestations. However, evidence now suggests a higher incidence of depressive manifestations in JHS/hEDS patients [9]. A meta-analysis also concluded that people with JHS/hEDS often present with more anxiety and depressive clinical manifestations [85]. This patient was diagnosed with a major depressive disorder. Research to establish that movement disorders are collateral of hEDS is not available yet [86]. However, it now is suggested that movement evasion might have developed to avoid pain, resulting in fixed dystonia [7], which was also present in our patient. Migraine could also be a clinical manifestation [87]. It has an earlier onset [88] and can reduce the patient’s quality of life. Our patient was treated for migraine and also referred to having a clinically disabling chronic headache. Even though the patient had relatively common clinical manifestations, she also had unusual ones. For instance, glaucoma has been reported in association with EDS [89]. Our patient also has glaucoma and a slight decrease in the anteroposterior diameter of the eyes. Although hEDS is associated with different rheumatic diseases [90], a few isolated patients have reported having costochondritis, as in this lady. However, no study has established this association yet. Additionally, the patient suffered from recurrent bilateral renal lithiasis diagnosed by the renal and urinary tract USG. She also underwent surgical interventions twice. Renal infarction could be present, but the association was previously established for vEDS [91]. Evidence supports the presence of gynecologic disorders, such as incontinence, endometriosis (also present in this patient), dyspareunia, etc., in EDS [92]. However, no association with uterine myomatosis was seen. A simple and contrast brain MRI revealed the presence of the right choroidal fissure cyst, which was considered a significant factor in EDS. A cyst has been reported in two patients, but they were meningeal cysts [93,94] and also have a low frequency in JHS/EDS-HT [95] (Appendix A).

The COL12A1 gene (NCBI ID: 1303) has 67 exons that extend over 121.937 base pairs and that are located in the region 6q13–q14.1. This gene encodes the alpha chain of type XII collagen (Uniprot ID: Q99715), a member of the fibril-associated collagens with interrupted triple helices (FACIT) collagen family. Type XII collagen is a homotrimer found in association with type I collagen. This association could modify the interactions between collagen I fibrils and the surrounding matrix. Genetic variants in this gene have been associated with myopathic Ehlers-Danlos syndrome (ORPHA:536516), Bethlem myopathy (ORPHA:610), and Ullrich congenital muscular dystrophy (ORPHA:75840) based on the information recorded in the ORPHANET [96], Online Mendelian Inheritance in Man (OMIM) (OMIM ID: 120320) [97], U.S. National Library of Medicine (NIH) [98], and Human Gene (GeneCards) (GCID: GC06M075084) [99] databases. To date, heterozygous and homozygous variants in the COL12A1 gene can cause mEDS. [100,101,102]. In this study, a heterozygous variant in the index case was evaluated. This variant replaced threonine with methionine at codon 2618 of the COL12A1 protein (p. Thr2618Met). The threonine residue was highly conserved, and a moderate physicochemical difference between threonine and methionine was seen. Threonine has a hydrophilic side chain, while methionine is a hydrophobic amino acid. This variant could alter the hydrophobicity index around it, which might influence the structure and function of the protein. However, additional studies should be conducted to determine the effect of this variant on the protein and confirm its possible association with the phenotype. This variant is located in the laminin G-like domain (amino acids 2520-2712), which binds to a polysaccharide of glucuronic acid–xylose units [103]. Laminin-G-containing proteins appear to play roles in cell adhesion, signaling, migration, assembly, and differentiation, based on the InterPro database (IPR001791). Patients with heterozygous mutations in the COL12A1 gene generally reveal a mild phenotype, with milder hypotonia, motor delay, and hyperlaxity, among other features. This reflects that carriers could express a mild phenotype with features overlapping with EDS or congenital myopathies with very mild symptoms, as reflected in the clinical examination [102,104]. Homozygous patients for COL12A1 mutations have shown hypertrophic cardiomyopathy, thus explaining the mild cardiomyopathic symptoms in the proband [105]. A recent report of a patient with similar features carrying an uncertain significant mutation in the COL12A1 gene suggested that this mutation might be important for the phenotype given that the patient has no pathogenic mutations in six collagen-related genes [106]. The clinical reports of patients with mutations in the COL12A1 were in agreement with congenital myopathy. However, it could not be assured that the diagnosis was of a certain type of EDS. The researchers usually describe the symptoms and keep with a diagnosis of mixed myopathy/congenital myopathy/EDS-like disease because the patients suffer from mild symptoms of the classic features of this type of disease, such as muscle hypotonia, joint contractures, joint hypermobility, skin abnormalities, facial dysmorphology, and developmental delay [102]. On the contrary, the variant found in this study shows a likely pathogenic effect at the protein level. On application of the ACMG guideline, PM1, PM2, PP2, PP3, and PP5 are enough for the likely pathogenic classification [61]. These features of the variant were provided by Varsome and Wintervar.

Traumatic brain injury and some of the findings of MRI could be associated with neuropsychological alterations in attention, memory, and executive functioning. Persistent headaches might also have affected the performance during the test. A characteristic of ED syndrome is small fiber neuropathy [107]. Cognitive deficits have been reported in this type of neuropathy [108] probably due to pain. Regardless of the cause of the neuropsychological alterations, they coincide with previous reports about deficits in concentration [10] and cognitive impairment in attention and memory [11]. The results of the emotional and personality tests reveal an emotionally stable, calm, and conscious person. It is not possible, in the current state of knowledge, to know whether these psychological characteristics are associated with the syndrome or if they are simply idiosyncratic to the patient.

Taking into consideration that there is a high degree of overlap between myopathic and connective tissue disorders [109], the prognosis in these patients is very variable. It has been proven that generated COL12A1 mutations in mice revealed fragile bones with disorganized collagen fiber arrangement, decreased expression of bone matrix proteins, and decreased bone-forming activity associated with delayed terminal differentiation [110]. Furthermore, another study found that the knockout mice had decreased grip strength, a delay in fiber-type transition, and a deficiency in passive force generation, while the muscle seemed more resistant to eccentric contraction-induced force drop, thus indicating a role for a matrix-based passive force-transducing elastic element in the generation of the weakness [100]. Hence, in the long term, the muscle weakness for the atrophy produced by this disease will be the main effect. 

## 4. Conclusions

In conclusion, a case of EDS is presented with a history of clinical manifestations that were a challenge for making an accurate diagnosis. Some of the clinical features were common, such as joint hypermobility, chronic pain, and hernias. However, other clinical features, such as costochondritis, spontaneous mandibular dislocation, and uterine myomatosis, were not at all common, lasting for more than 40 years of evolution. 

The classification of this patient was challenging, but the molecular analysis assisted in the process. The criteria for mEDS include inheritance, which can be autosomal dominant or autosomal recessive. The major criteria included the following: congenital muscle hypotonia and/or muscle atrophy that improved with age, proximal joint contractures (knee, hip, and elbow), and hypermobility of distal joints, and the minor criteria included the presence of soft, doughy skin and atrophic scarring. Some of these clinical manifestations were presented by our patient. Numerous EDS clinical manifestations have overlapping presentations. Hence, the prognosis cannot be defined. The therapeutic proposals for this specific type of EDS are not directed to the type per se. Hence, the focus is on pain management [36] and physical rehabilitation. 

## Figures and Tables

**Figure 1 genes-13-02118-f001:**
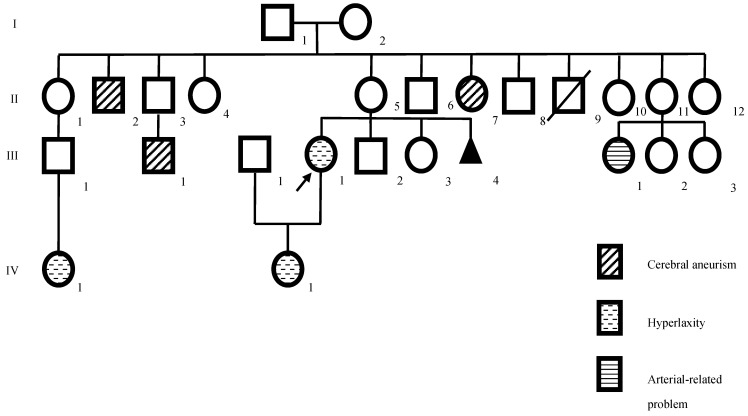
The family pedigree of the patient (III-1), reveals related diseases in the patient family, like a history of cerebral aneurism, hyperlaxity, and diseases in the blood vessels.

**Figure 2 genes-13-02118-f002:**
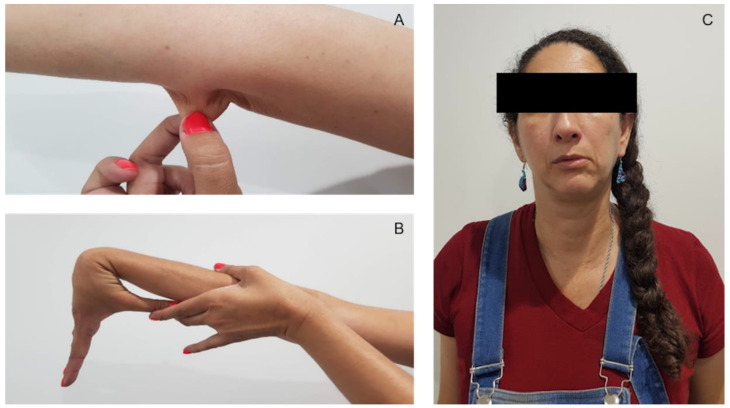
Clinical features of the patient. (**A**) Hyperextensibility of the skin. (**B**) Passive apposition of the thumbs to the flexor aspects of the forearms. (**C**) Mandibular deviation to the left.

**Figure 3 genes-13-02118-f003:**
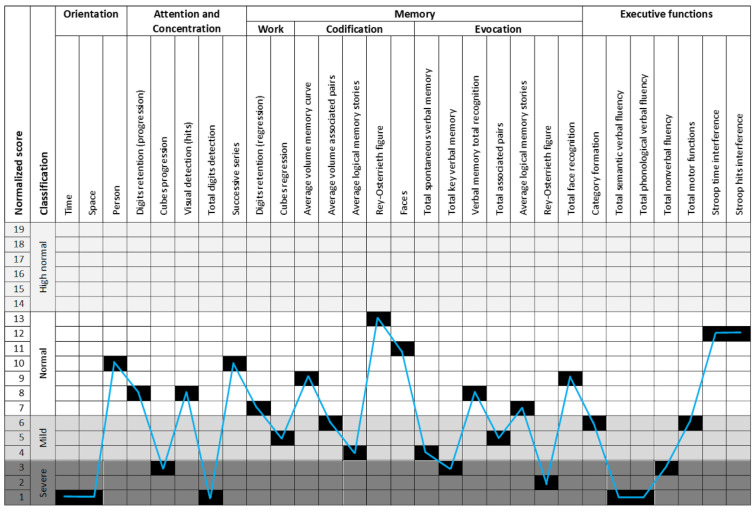
Neuropsi test results. The 26 tasks included assessing each neuropsychological function (attention, memory, and executive functions). Most of the results revealed mild and severe alterations. Some others were at a low normal level.

**Table 1 genes-13-02118-t001:** The Beighton Hypermobility Score.

Clinical Finding/Maneuver	Score	Patient
Hyperextension of the elbows beyond 10°.		
Right	1	1
Left	1	1
Hyperextension of the knees beyond 10°.		
Right	1	1
Left	1	1
Passive apposition of the thumbs to the flexor aspects of the forearms.		
Right	1	1
Left	1	1
Passive dorsiflexion of the little fingers beyond 90°.		
Right	1	0
Left	1	0
Forward flexion of the trunk, with knees fully extended, so that the palms of the hands rest easily on the floor.	1	0
Maximum score	9	6

**Table 2 genes-13-02118-t002:** Ehlers Danlos-2017 Classification.

Clinical Subtype	Abbreviation	Old Nomenclature	Inheritance Pattern	Gene	Protein	Predominant Clinical Features
Classical EDS	cEDS	Classical EDS, types I and II	AD	*COL5A1*, *COL5A2*, rarely *COL1A1*	Type V collagen Type I collagen	Hyperextensible skin, atrophic scars, fragile skin, increased bruisability, doughy/velvety skin, and generalized joint hypermobility
Classical-like EDS	clEDS	TNXB-deficient EDS	AR	*TNXB*	Tenascin XB	Hyperextensible skin, velvety skin texture, no atrophic scarring, generalized joint hypermobility, and easy bruisability
Cardiac-valvular	cvEDS		AR	*COL1A1*	Type I collagen	Progressive cardiac valve involvement, hyperextensible thin skin, atrophic scars, increased bruisability, and joint hypermobility
Vascular EDS	vEDS	Vascular EDS, type IV	AD	*COL3A1*	Type III collagen Type I collagen	Arterial rupture, internal organ rupture (colon, uterus), severe bruising, thin translucent skin, and small joint hypermobility
Hypermobile EDS	hEDS	Hypermobile EDS, type III	AD	Unknown	Unknown	Generalized joint hypermobility, mildly hyperextensible skin, soft velvety skin, recurrent hernias, organ prolapse, unexplained striae, chronic pain, and joint dislocations/subluxations
Arthrochalasia EDS	aEDS	Arthrochalasia, types VIIA and VIIB	AD	*COL1A1*, *COL1A2*	Type I collagen	Congenital bilateral hip dislocation, severe generalized joint hypermobility, hyperextensible skin, tissue fragility, hypotonia, and mild osteopenia
Dermatosparaxis EDS	dEDS	Dermatosparaxis EDS, type VIIC	AR	*ADAMTS2*	ADAMTS-2	Severe skin fragility, visceral fragility, lax redundant skin, severe bruisability, and postnatal growth retardation
Kyphoscoliotic EDS	kEDS	Kyphoscoliosis EDS, type VI	AR	*PLOD1* *FKBP14*	LH1 FKBP22	Congenital hypotonia, early kyphoscoliosis, generalized joint hypermobility, osteopenia, blue sclerae, and marfanoid habitus
Brittle Cornea Syndrome	BCS	Brittle cornea syndrome	AR	*ZNF469* *PRDM5*	ZNF469 PRDM5	Thin cornea, keratoconus, blue sclerae, the risk for globe rupture, retinal detachment, and high myopia
Spondylodysplastic EDS	spEDS	EDS progeroid type Spondylocheirodysplastic EDS	AR	*B4GALT7* *B3GALT6* *SLC39A13*	β4GalT7 β3GalT6 ZIP13	Short stature, hypotonia, limb bowing, characteristic skeletal findings, osteopenia, hyperextensible, and thin doughy skin
Musculocontractural EDS	mcEDS	Adducted thumb Clubfoot Syndrome B3GalT6-deficient EDS EDS Kosho type	AR	*CHST14* *DSE*	D4ST1 DSE	Congenital contractures (clubfoot), hyperextensible skin, easy bruisability, fragile skin, atrophic scars, and recurrent dislocations
Myopathic EDS	mEDS		AD or AR	*COL12A1*	Type XII collagen	Congenital hypotonia, proximal joint contractures, distal joint hypermobility, doughy skin, and atrophic scars
Periodontal EDS	pEDS	EDS periodontitis, type VIII	AD	*C1R* *C1S*	C1r C1s	Early-onset severe periodontitis, unattached gingiva, pretibial plaques, joint hypermobility, hyperextensible skin, and marfanoid features

AD, autosomal dominant; AR, autosomal recessive.

**Table 3 genes-13-02118-t003:** The Five-Point Questionnaire.

The Five-Point Questionnaire
1. Can you now (or could you ever) place your hands flat on the floor without bending your knees?
2. Can you now (or could you ever) bend your thumb to touch your forearm?
3. As a child, did you amuse your friends by contorting your body into strange shapes or could you do the splits?
4. As a child or teenager, did your shoulder or kneecap dislocate on more than one occasion?
5. Do you consider yourself “double-jointed”?
A “yes” answer to two or more questions suggests joint hypermobility with 80–85% sensitivity and 80–90% specificity

**Table 4 genes-13-02118-t004:** Description of a variant candidate identified with the ANNOVAR tool in the index case with Ehlers-Danlos syndrome. Chr: Chromosome, Ref: Reference allele, Alt: Alternate allele, dbSNP: NCBI SNP database, Change: nucleic acid/amino acid change, 1000G: 1000 genomes database, ExAC: Exome Aggregation Consortium, gnomAD: genome aggregation database, SIFT: Sorting Intolerant from Tolerant tool, PolyPhen2: Polymorphism Phenotyping v2 tool, PROVEAN: Protein Variation Effect Analyzer tool, CADD: Combined Annotation Dependent Depletion tool. D: Deleterious. GT: Genotype.

Genetic Variant Information						Allele Frequency			Pathogenicity Predictors				GT
Gene	Chr	Ref	Alt	dbSNP	Change	1000G	ExAC	gnomAD	SIFT	PolyPhen2	PROVEAN	CADD	
COL12A1	6	C	T	rs201988277	exon51 c.C7853T p. T2618M	0.0002	0.0003	0.0003	D	D	D	25.2	0/1

## Data Availability

The data presented in this study are available on request from the corresponding author.

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
