# Peer review of "Ehlers-Danlos: A Literature Review and Case Report in a Colombian Woman with Multiple Comorbidities"

_genes, 2022, doi:10.3390/genes13112118_

Round 1
Reviewer 1 Report
The article is an interesting description of a patient with myopathic type of Ehlers-Danlos syndrome (EDS) with a very long lasting diagnostic odyssey from the first symptoms of the disease to the adequate molecular diagnosis of EDS. The article contains also a basic information about the syndrome, its types, causal mutations and patients treatment and care.
I have not important substantive remarks. However, the whole text isn't well written, and English language requires extensive correction.
Author Response
Thank you for the opportunity to revise and re-submit our manuscript. We are grateful for the expert review and feel we have significantly improved the paper. We thank the reviewer for her/his observations. Considering the comments of the reviewer, academic peers (,:Enago Copyediting service) were consulted to improve the English edition.

Reviewer 2 Report
It is very beneficial to report patients with well described genetic disorders that do not exactly fit the commonly accepted criteria. Every day we find exceptions and variability and these cases are important report to expand general knowledge and hope it leads to quicker diagnosis for patients with non-classic symptoms. Well done.
Thank being said, although the English is correct, in many cases the grammar suffers and the subject of the sentence not clear and often misleading.
Line 17, the past tense should be used "took"
Line 74 and 75 is very poorly written with too many subjects. Should be two sentences and changed to "is, therefore,"
Line 129-130, "being the heart" should not be used when referring to other anatomical structures
Line 137 needs work
Line 298 "patients have notified a delay" is nonsensical
Thus, one more pass of the manuscript with an eye for grammar would take this from a good paper to an excellent paper.
Author Response
Thank you for the opportunity to revise and re-submit our manuscript. We are grateful for the expert review and feel we have significantly improved the paper. We thank the reviewer for her/his observations. Following the reviewer's suggestion, academic peers (,:Enago Copyediting service) were consulted to improve the English edition and all typographical errors have been amended.

Round 2
Reviewer 1 Report
I have no comments for the Authors. The present form of the article is much better than the previous one.